# Isotherm Theoretical Study of the Al_x_Ga_1−x_As_y_Sb_1−y_ Quaternary Alloy Using the Regular Solution Approximation for Its Possible Growth via Liquid-Phase Epitaxy at Low Temperatures

**DOI:** 10.3390/e24121711

**Published:** 2022-11-23

**Authors:** Erick Gastellóu, Rafael García, Ana M. Herrera, Antonio Ramos, Godofredo García, Mario Robles, Jorge A. Rodríguez, Yani D. Ramírez, Roberto C. Carrillo

**Affiliations:** 1División de Sistemas Automotrices, Universidad Tecnológica de Puebla (UTP), Puebla C.P. 72300, Puebla, Mexico; 2Departamento de Investigación en Física, Universidad de Sonora (UNISON), Hermosillo C.P. 83000, Sonora, Mexico; 3Centro de Investigación en Dispositivos Semiconductores, Benemérita Universidad Autónoma de Puebla (BUAP), Puebla C.P. 72570, Puebla, Mexico; 4Universidad Tecnológica de Puebla (UTP), Puebla C.P. 72300, Puebla, Mexico; 5Departamento de Física, Universidad de Sonora (UNISON), Hermosillo C.P. 83000, Sonora, Mexico

**Keywords:** regular solution approximation, isotherm diagram, semiconductor compound, LPE

## Abstract

This work presents the theoretical calculation of isotherm diagrams for quaternary alloys of III–V semiconductor compounds with the form III_x_III_1−x_V_y_V_1−y_. In particular, the isotherm diagrams for the Al_x_Ga_1−x_As_y_Sb_1−y_ quaternary alloy at low temperatures were calculated (500 °C, 450 °C, 400 °C, and 350 °C). The Al_x_Ga_1−x_As_y_Sb_1−y_ quaternary alloy was formed from four binary compounds such as GaAs, AlAs, AlSb, and GaSb, all with direct bandgaps. The regular solution approximation was used to find the quaternary isotherm diagrams, represented in four linearly independent equations, which were solved using Parametric Technology Corporation Mathcad 14.0 software for different arsenic and antimony atomic fractions. The results support the possible growth of layers via liquid-phase epitaxy in a range of temperatures from 500 °C to 350 °C, where the crystalline quality could be improved at low temperatures. These semiconductor layers could have applications for optoelectronic devices in photonic communications, thermophotovoltaic systems, and microwave devices with good crystalline quality.

## 1. Introduction

The world is currently living a technological revolution in electronic communications, where silicon has been a fundamental aspect of technology for a long time. However, when the operation of a laser diode was demonstrated in 1960 [1], direct bandgap semiconductors obtained great importance. These materials are III–V semiconductor compounds, which can form binary, ternary, and quaternary alloys, whose structural, electric, and optical properties have applications in devices such as light-emitting diodes, laser diodes, electro-optical switches, photodetectors, high-mobility transistors, microwave devices, solar cells, thermophotovoltaic systems, ultrahigh-power devices, and heterostructure devices [2,3,4]. GaAs, GaN, GaSb, InN, and AlN are some of the most important binary compounds, while AlGaAs, AlGaSb, AlGaN, and InGaN are ternary compounds with applications in photonics devices. Furthermore, InAlGaN, InGaAsSb, and InGaAsP are some of the quaternary alloys most important to optoelectronic and thermophotovoltaic systems [5].

On the other hand, III–V semiconductor compounds have generally been obtained using methods such as MOCVD (metal–organic chemical vapor deposition), MBE (molecular beam epitaxy), and LPE (liquid-phase epitaxy). MOCVD and MBE might have some disadvantages due to the use of highly toxic gases and high operating costs in their growth systems, respectively. LPE is an easy method to operate, with low cost, and it is widely used for the growth of heterostructure semiconductor devices [5,6]. Avalanche photodiodes of Al_x_Ga_1−x_Sb alloy were obtained via LPE, which were tuned with a wavelength of 1.55 µm (0.80 eV), where the silica fibers have their lowest losses [7]. Figure 1 shows the components and chamber of a liquid-phase epitaxy system [2]. The lower part of Figure 1a shows a graphite boat, in which it is possible to observe the “niches” where the metallic solids that make up the mixture are placed, while the upper part shows the cavities for substrates. Figure 1b shows the sliding graphite boat and its manipulator on the quartz support, where the mixture and the substrate do not have contact at the beginning of the growth. However, when the system reaches the liquidus temperature using the quartz manipulator, the upper rule is slipped, initiating contact between the mixture and the substrate. At this moment, the temperature is slightly lowered to break the equilibrium of the solution and deposit the excess solution on the substrate (depending on the growth technique). On the other hand, Figure 1c shows the chamber of the liquid-phase epitaxy system, where the heater covered by a layer of gold can be observed to concentrate the emitted radiation and generate a flat area large enough to not have temperature gradients during growth. It is important to mention that the processes in LPE growth are carried out in an ultrahigh-purity hydrogen environment.

In particular, Al_x_Ga_1−x_As_y_Sb_1−y_ quaternary alloy lattice-matched to GaSb substrates is an attractive material in optoelectronic devices such as photodetectors and thermophotovoltaic systems [5]. The importance of this material is due to its wide bandgap as a function of the x–y composition (1.0–1.8 µm in wavelength), which was previously grown via LPE at 605 °C [8]. In another study [9], a type p Al_0.6_Ga_0.4_As_0.05_Sb_0.9_ layer lattice-matched to GaSb substrates was also grown via LPE at 635 °C to obtain a p–n junction. Toginho et al. reported a study of the photoluminescence properties in undoped and Te-doped AlGaAsSb alloys synthesized by MBE [10]. On the other hand, Hassan et al. employed density functional theory (DFT) to study the structural, electronic, optical, and thermodynamic properties of Al_x_Ga_1−x_As_y_Sb_1−y_ quaternary alloys [11], while Pessetto et al. calculated the phase diagrams using the delta lattice parameter (DLP) model to grow the Al_x_Ga_1−x_As_y_Sb_1−y_ quaternary alloy via LPE at temperatures above 700 °C [12]. However, Ito showed that the essence of the DLP model uses the bonding energy proportional to *a*^−2.5^ (*a* = lattice parameter) in semiconductors to calculate the mixing enthalpies with good results, principally in A_x_B_1−x_C ternary alloys with *x* = 0.5 [13]. Then, different studies on Al_x_Ga_1−x_As_y_Sb_1−y_ quaternary alloy showed growth temperatures above 500 °C, which could produce structural defects during layer deposition at high temperatures. This work presents the isotherm diagrams obtained for Al_x_Ga_1−x_As_y_Sb_1−y_ quaternary alloy, indicating the possible growth of layers via liquid-phase epitaxy lattice-matched to GaSb substrates at temperatures below 500 °C. The theoretical analysis of isotherm diagrams was conducted using the regular solution approximation, which has been extensively studied and proven in quaternary alloys such as AlGaPAs [14,15]. The possible obtention of AlGaAsSb layers at temperatures below 500 °C could enhance the crystalline quality during the growth, thereby decreasing structural defects. Another application of AlGaAsSb layers with a good crystalline quality might be in their use as substrates or buffer layers for the investigation of nitride (III) materials via nitridation [16].

## 2. Theoretical Model

One of the most important thermodynamic models to calculate isotherm diagrams for a quaternary system is the regular solution approximation proposed by Jordan and Ilegems [14]. This thermodynamic model is based on the solid–liquid equilibrium in solid solutions for a substitutional model, in which an atom of one type can substitute another inside of the same sublattice; in this case, there are two sublattices (cation–anion) with four binary compounds. The isotherm diagrams for a quaternary system can be used to calculate the weights for layer process growth via LPE. 

It is important to mention that the equations presented below were obtained from previous studies [14,15], in which the conceptual and mathematical development was presented. The theoretical study begins with the solid–liquid equilibrium, where the Gibbs free energy per mole $G_{m}$ of a quaternary solid in equilibrium with a quaternary liquid phase is given by the following equation:(1)$$G_{m}=2\overset{4}{\underset{i=1}{\sum }}x_{i}\mu_{i}=2\overset{4}{\underset{i=1}{\sum }}x_{i}\mu_{i}^{l}.$$
here $\mu_{i}$ and $\mu_{i}^{l}$ are the chemical potentials of elemental component *i* (*i* = 1, 2, 3, 4), in the solid and liquid phase. Furthermore, *x_i_* is the atom fraction of component *i* in the solid [14]. The chemical potentials ($\mu_{i}^{l}$) can be expressed as a function of temperature and total pressure, which is given by the following equation:(2)$$\mu_{i}^{l}=\mu_{i}^{0l}+RT\ln a_{i}^{l}.$$
here $\mu_{i}^{0l}$ is the standard chemical potential in the liquid (J·mol^−1^), R the universal gas constant (8.314472 J·K^−1^·mol^−1^), T is the temperature (K), and $a_{i}^{l}$ is the activity of each element in the liquid.

Then, the Gibbs free energy for a quaternary solid solution according to Equation (1) can be expressed as follows:(3)$$G_{m}=x\mu_{1}+\left(1-x\right)\mu_{2}+y\mu_{3}+\left(1-y\right)\mu_{4}.$$
here, in the liquid, there are four chemical potentials ($\mu_{1},\;\mu_{2},\;\mu_{3},\;\mu_{4}$), corresponding to the four binary compounds AC, AD, BC, and BD, which, when multiplied by their respective molar fractions $x_{13}$, $x_{14}$, $x_{23}$, and $x_{24}$, results in the following equation:(4)$$\;G_{m}=x_{13}\mu_{13}+x_{14}\mu_{14}+x_{23}\mu_{23}+x_{24}\mu_{24}.$$

From the mass balance, the relationships of x, y, and x_ij_ can be obtained; therefore, Equation (4) can be represented in the following form:(5)$$\;G_{m}=\left(x+y-1\right)\left(\mu_{1}+\mu_{3}\right)+\left(1-y\right)\left(\mu_{1}+\mu_{4}\right)+\left(1-x\right)\left(\mu_{2}+\mu_{3}\right).$$
here the chemical potentials of the four binary compounds must satisfy the following condition:AD+BC=AC+BD.

Then, the expression for the Gibbs free energy of mixing from four pure binary compounds, $G^{M}$, derived by Jordan and Ilegems [15], can be obtained as follows:(6)$$G^{M}=RT\left(n_{1}\ln\frac{n_{1}}{n_{1}+n_{2}}+n_{2}\;\ln\frac{n_{2}}{n_{1}+n_{2}}+n_{3}\;\ln\frac{n_{3}}{n_{3}+n_{4}}+n_{4}\;\ln\frac{n_{4}}{n_{3}+n_{4}}\right)+\;\alpha_{34}\;\frac{n_{3}n_{4}}{n_{3}+n_{4}}+\alpha_{c}\;\frac{n_{14}n_{23}-n_{13}n_{24}}{n_{1}+n_{2}}.$$
here the mass balances denoted by $n_{1}=n_{13}+n_{14}$, $n_{2}=n_{23}+n_{24}$, $n_{3}=n_{13}+n_{23}$, and $n_{4}=n_{14}+n_{24}$ are the number of moles of elemental components, whereas $n_{13}$, $n_{14}$, $n_{23}$, and $n_{24}$ are the number of moles of the binary compound components forming the quaternary solid solution. Furthermore, $\alpha_{c}=\alpha_{13}+\alpha_{24}-\alpha_{23}-\alpha_{14}$, binary, and ternary interaction parameters are derived from experimental data [8,14,17]. Then, using the partial derivatives of Equations (2) and (6) for the ij elements, the following equation can be obtained:(7)$$\frac{\partial G^{M}}{\partial n_{ij}}=RT\ln a_{ij}.$$

We can obtain the explicit relationships for $\alpha_{13},\;\alpha_{24},\;\alpha_{23},\;\mathrm{and}\;\alpha_{14}$ through the following equations: (8a)$$RT\ln a_{13}=RT\ln xy+\alpha_{12}{\left(1-x\right)}^{2}+\alpha_{34}{\left(1-y\right)}^{2},$$
(8b)$$RT\ln a_{14}=RT\ln X\left(1-y\right)+\alpha_{12}{\left(1-x\right)}^{2}+\alpha_{34}y^{2}+\alpha_{c}\left(1-x\right)y,$$
(8c)$$RT\ln a_{23}=RT\ln\left(1-x\right)y+\alpha_{12}x^{2}+\alpha_{34}{\left(1-y\right)}^{2}+\alpha_{c}\left(1-y\right)x,$$
(8d)$$RT\ln a_{24}=RT\ln\left(1-x\right)\left(1-y\right)+\alpha_{12}x^{2}+\alpha_{34}y^{2}+\alpha_{c}xy.$$

Therefore, the explicit equations of the solid–liquid equilibrium for the form III_x_III_1−x_V_y_V_1−y_ were obtained, which were used in particular for the Al_x_Ga_1−x_As_y_Sb_1−y_ quaternary alloy to evaluate its possible growth via LPE at low temperatures, according to the following equations:(9)$$\begin{array}{l} \Delta H_{F}^{13}-T\Delta S_{F}^{13}-RT\ln xy+RT\ln4x_{1}^{l}\;x_{3}^{l}=M_{13}^{l}+\alpha_{12}{\left(1-x\right)}^{2}+\alpha_{34}{\left(1-y\right)}^{2}+\\ \;+\alpha_{c}\left(1-x\right)\left(1-y\right), \end{array}$$
(10)$$\begin{array}{l} \Delta H_{F}^{14}-T\Delta S_{F}^{14}-RT\ln x\left(1-y\right)+RT\ln4x_{1}^{l}\;x_{4}^{l}=M_{14}^{l}+\alpha_{12}{\left(1-x\right)}^{2}+\\ \;+\alpha_{34}y^{2}+\alpha_{c}\left(1-x\right)y, \end{array}$$
(11)$$\begin{array}{l} \Delta H_{F}^{23}-T\Delta S_{F}^{23}-RT\ln\left(1-x\right)y+RT\ln4x_{2}^{l}\;x_{3}^{l}=M_{23}^{l}+\alpha_{12}x^{2}+\alpha_{34}{\left(1-y\right)}^{2}+\\ \;+\alpha_{c}x\left(1-y\right), \end{array}$$
(12)$$\begin{array}{l} \Delta H_{F}^{24}-T\Delta S_{F}^{24}-RT\ln\left(1-x\right)\left(1-y\right)+RT\ln4x_{2}^{l}\;x_{4}^{l}=M_{24}^{l}+\alpha_{12}x^{2}+\\ \;+\alpha_{34}y^{2}-\alpha_{c}xy. \end{array}$$
here $\alpha_{c}$ has the following form:$$\alpha_{c}=\Delta H_{F}^{14}-T\Delta S_{F}^{14}+\Delta H_{F}^{23}-T\Delta S_{F}^{23}-\Delta H_{F}^{13}+T\Delta S_{F}^{13}-\Delta H_{F}^{24}+T\Delta S_{F}^{24}+\frac{1}{2}\left(\alpha_{13}^{l}+\alpha_{24}^{l}-\alpha_{23}^{l}-\alpha_{14}^{l}\right).$$
And.
$$M_{ij}^{l}=\alpha_{ij}^{l}\left[0.5-x_{i}^{l}\;\left(1-x_{j}^{l}\right)-x_{j}^{l}\left(1-x_{i}^{l}\right)\right]+\left(\alpha_{ik}^{l}x_{k}^{l}+\alpha_{im}^{l}x_{m}^{l}\right)\left(2x_{i}^{l}-1\right)+\;\left(\alpha_{jk}^{l}x_{k}^{l}+\alpha_{jm}^{l}x_{m}^{l}\right)\left(2x_{l}^{l}-1\right)+2\alpha_{km}^{l}x_{k}^{l}x_{m}^{l},\;i,j,k,m=1,\;4;\;\;\;\;\;\;\;\;\;\;\;\;\;\;\;\;\;i\neq j\neq m\neq k.$$

Moreover, $\alpha_{ij}^{l}$ denotes the six interaction energies in the quaternary solid solution.

Considering that the fusion enthalpy can be written in terms of the fusion entropy $\left(\Delta H_{F}=\Delta S_{F}T_{F}\right)$, Equations (9)–(12) can be written as follows:(9a)$$\begin{array}{l} \Delta S_{F}^{13}\left(T_{F}^{13}-T\right)-RT\ln xy+RT\ln4x_{1}^{l}\;x_{3}^{l}=M_{13}^{l}+\alpha_{12}{\left(1-x\right)}^{2}+\alpha_{34}{\left(1-y\right)}^{2}+\\ \;+\alpha_{c}\left(1-x\right), \end{array}$$
(10a)$$\begin{array}{l} \Delta S_{F}^{14}\left(T_{F}^{14}-T\right)-RT\ln x\left(1-y\right)+RT\ln4x_{1}^{l}\;x_{4}^{l}=M_{14}^{l}+\alpha_{12}{\left(1-x\right)}^{2}+\\ \;+\alpha_{34}y^{2}+\alpha_{c}\left(1-x\right)y, \end{array}$$
(11a)$$\begin{array}{l} \Delta S_{F}^{23}\left(T_{F}^{23}-T\right)-RT\ln\left(1-x\right)y+RT\ln4x_{2}^{l}\;x_{3}^{l}=M_{23}^{l}+\alpha_{12}x^{2}+\alpha_{34}{\left(1-y\right)}^{2}+\\ \;+\alpha_{c}x\left(1-y\right), \end{array}$$
(12a)$$\begin{array}{l} \Delta S_{F}^{24}\left(T_{F}^{24}-T\right)-RT\ln\left(1-x\right)\left(1-y\right)+RT\ln4x_{2}^{l}\;x_{4}^{l}=M_{24}^{l}+\alpha_{12}x^{2}+\\ \;+\alpha_{34}y^{2}-\alpha_{c}xy. \end{array}$$

Lastly, for the Al_x_Ga_1−x_As_y_Sb_1−y_ quaternary alloy, the interaction parameters for the binaries, fusion entropies, fusion temperatures, and interaction parameters of the ternaries were searched in the literature, assigning the conditions Al = 1, Ga = 2, As = 3, and Sb = 4 [14,15,17,18]. Therefore, Equations (9a)–(12a), which are linearly independent, were solved numerically using the PTC Mathcad 14.0 software. The numerical values of the parameters mentioned above are shown in Table 1. It is important to mention that, when the temperatures were entered into the software, a conversion from kelvin (K) to degrees Celsius (°C) was performed before plotting the graphs.

## 3. Results and Discussion

Figure 2 shows the quaternary alloy isotherm diagrams obtained at different growth temperatures (350 °C, 400 °C, 450 °C, and 500 °C). In Figure 2, the isotherms are shown as horizontal lines in which transformations took place between mixtures of various components with a minimum melting point (solidification), which was lower for each of the components in its pure state. This occurred in mixtures with high stability in the liquid state, whose components were insoluble in the solid state. In this study, the isotherms showed a relationship among the aluminum, arsenic, and antimony compositions in a gallium-rich zone as a function of different temperatures (350 °C, 400 °C, 450 °C, and 500 °C. Figure 2 shows four constant atomic fractions for aluminum (0.0041 mol, 0.0035 mol, 0.0025 mol, and 0.002 mol), whose values decreased when the growth temperature increased, as well as when the arsenic and antimony atomic fractions increased. It is also possible to observe a certain correspondence between the arsenic and antimony atomic fractions. Something very important to mention is that, when temperature increased, the aluminum atomic fraction decreased, and the arsenic atomic fraction increased from 10^−8^ for 350 °C to 10^−5^ for 500 °C. On the other hand, as the aluminum, arsenic, and antimony atomic fractions were small, the quaternary alloy was in the gallium-rich region as the solvent, while the other elements were solutes in the metallic solution. The aluminum fraction values in Figure 2 could indicate that, upon continuing to increase the saturation temperature of the Al_x_Ga_1−x_As_y_Sb_1−y_ quaternary alloy, the aluminum atomic fraction would continue to decrease; accordingly, the antimony and arsenic atomic fractions would continue to increase until reaching the GaAsSb ternary alloy. Furthermore, when the temperature decreased, the arsenic fraction also decreased until reaching the AlSbGa ternary alloy, whereas the GaAsSb ternary alloy could be obtained upon increasing the temperature.

In Figure 2, it can also be observed that, when the temperature increased in the range from 350 °C to 500 °C, the aluminum atomic fraction decreased from 0.0041 to 0.002. However, the arsenic and antimony atomic fractions increased. A similar behavior in phase diagrams was reported by Ilegems and Panish for the Al_x_Ga_1−x_P_y_As_1−y_ quaternary alloy (see graph 1 in their study) [15]. This behavior agrees with the results obtained in our study, suggesting their accuracy. 

Figure 3 shows the atomic fractions in solid x and y for the quaternary alloy, as a function of the arsenic atomic fraction in the liquid ($x_{As}^{l}$) for the temperatures of 500 °C, 450 °C, 400 °C, and 350 °C. It is possible to observe a relationship between Figure 2 and Figure 3, as a function of the arsenic atomic fraction in the liquid ($x_{As}^{l}$), where it can be related in Figure 2 to the atomic fractions of aluminum and antimony in the liquid ($x_{Al}^{l}$, and $x_{Sb}^{l}$) for the different temperatures. Figure 3a shows that the atomic fractions in solid x and y converged to 0.1 for the arsenic atomic fraction in the liquid with a value of 1 × 10^−4^ mol, along with an aluminum atomic fraction in the liquid of 0.002 mol, and an antimony atomic fraction in the liquid of approximately 0.03 mol (in the gallium-rich region). Figure 3b shows that the atomic fractions in solid x and y began to separate, shifting toward smaller values of the arsenic atomic fraction in the liquid (approximately 10^−5^), with values for the atomic fractions of aluminum and antimony in the liquid of 0.0025 mol and 0.01 mol (450 °C), respectively. Figure 3c shows a greater separation between the atomic fractions in solid x and y, for an approximate value of 10^−6^ mol for the arsenic atomic fraction in the liquid, with values for the atomic fractions of aluminum and antimony of 0.0035 mol and 0.004 mol, respectively, in the liquid at 400 °C. Lastly, Figure 3d shows the maximum separation between the atomic fractions in solid x and y, for a value of approximately 10^−7^ mol for the arsenic atomic fraction in the liquid, with values for the atomic fractions of aluminum and antimony of 0.0041 mol and approximately 0.001 mol (in the gallium-rich region), respectively, in the liquid at 350 °C.

Therefore, it was observed again that, when the temperature decreased, the arsenic atomic fraction continued to decrease, approximating the AlGaAsSb quaternary alloy to the AlGaSb ternary alloy, as mentioned in Figure 2.

## 4. Conclusions

Isotherm diagrams of the Al_x_Ga_1−x_As_y_Sb_1−y_ quaternary alloy were calculated using a regular solution approximation for possible growth of layers at temperatures below 500 °C, via liquid-phase epitaxy. A similar behavior in Al_x_Ga_1−x_P_y_As_1−y_ quaternary alloy has been reported, agreeing with the results obtained in our study and suggesting their accuracy. Isotherm diagrams showed that, when the temperature increased from 350 °C to 500 °C, the aluminum atomic fraction decreased from 0.0041 mol to 0.002 mol, whereas the arsenic and antimony atomic fractions increased. On the other hand, graphs of the atomic fractions in solids x and y, as a function of the arsenic atomic fraction in the liquid, showed that, at low temperatures, the x fraction increased, while the y fraction decreased. Furthermore, the arsenic atomic fraction maintained a very small value until reaching the AlGaSb ternary alloy.

## Figures and Tables

**Figure 1 entropy-24-01711-f001:**
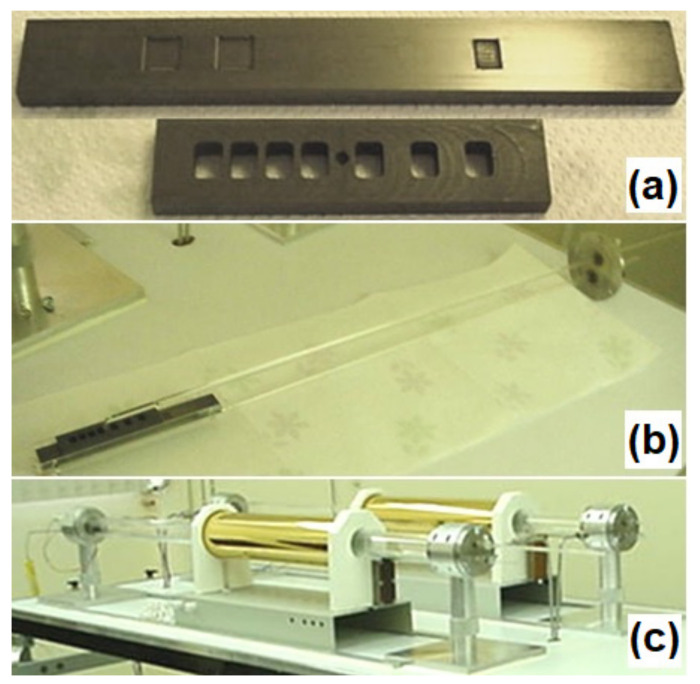
Liquid-phase epitaxy system: (**a**) sliding graphite boat; (**b**) quartz manipulator; (**c**) LPE system chamber.

**Figure 2 entropy-24-01711-f002:**
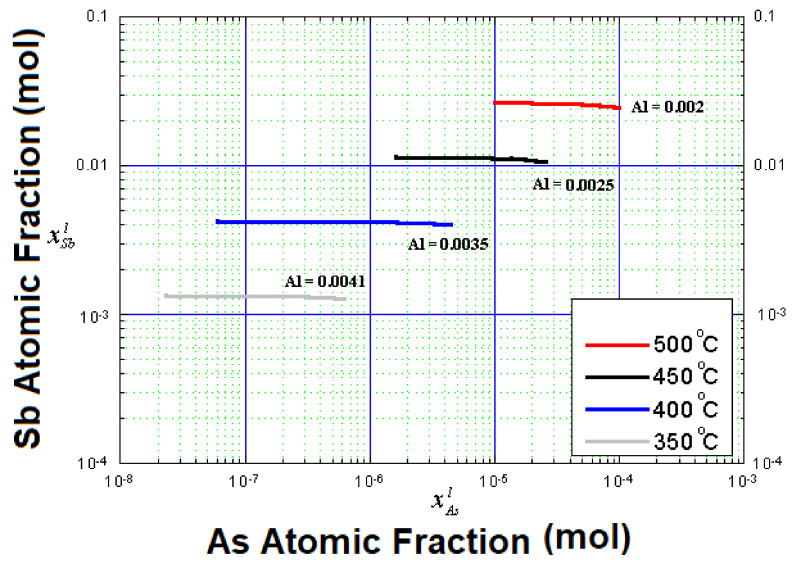
Isotherm diagrams of the Al_x_Ga_1−x_As_y_Sb_1−y_ quaternary alloy at low temperatures indicating its possible growth via liquid-phase epitaxy.

**Figure 3 entropy-24-01711-f003:**
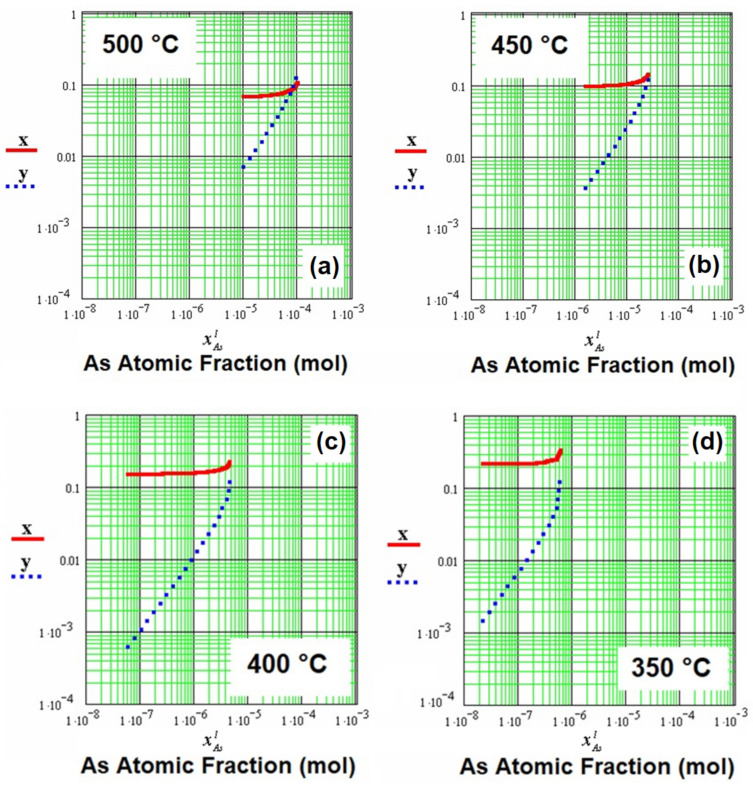
Atomic fractions in solids x and y as a function of the arsenic atomic fraction in the liquid $x_{As}^{l}$ for different temperatures: (**a**) 500 °C, (**b**) 450 °C, (**c**) 400 °C, and (**d**) 350 °C.

**Table 1 entropy-24-01711-t001:** Interaction parameters, fusion entropies, and fusion temperatures of binary compounds, along with interaction parameters for the ternary compounds [8].

Interaction Parameters in Liquid Phase (cal·mol^−1^)			Fusion Entropies (cal·mol^−1^·K^−1^)			Fusion Temperatures of Binary Compounds (K)			Interaction Parameters for the Ternary in Solid Phase (cal·mol^−1^)	
$\alpha_{12}\left(T\right)$	$\alpha_{AlGa}\left(T\right)$	104	$\Delta S_{\mathrm{AlAs}}^{F}$	$\delta_{13}$	15.6	$T_{\mathrm{AlAs}}^{F}$	$T_{13}$	2043	${\mathrm{AlAs}}_{x}{\mathrm{Sb}}_{1-x}$	4200
$\alpha_{13}\left(T\right)$	$\alpha_{AlAs}\left(T\right)$	−12T+600								
$\alpha_{14}\left(T\right)$	$\alpha_{AlSb}\left(T\right)$	4539–9.16T	$\Delta S_{\mathrm{AlSb}}^{F}$	$\delta_{14}$	14.74	$T_{\mathrm{AlSb}}^{F}$	$T_{14}$	1338	${\mathrm{GaAs}}_{x}{\mathrm{Sb}}_{1-x}$	4000
$\alpha_{23}\left(T\right)$	$\alpha_{GaAs}\left(T\right)$	5160–9.16T	$\Delta S_{\mathrm{GaAs}}^{F}$	$\delta_{23}$	16.64	$T_{\mathrm{GaAs}}^{F}$	$T_{23}$	1511	${\mathrm{Al}}_{x}{\mathrm{Ga}}_{1-x}\mathrm{As}$	0
$\alpha_{24}\left(T\right)$	$\alpha_{GaSb}\left(T\right)$	4700–6T	$\Delta S_{\mathrm{GaSb}}^{F}$	$\delta_{24}$	15.8	$T_{\mathrm{GaSb}}^{F}$	$T_{24}$	985	${\mathrm{Al}}_{x}{\mathrm{Ga}}_{1-x}\mathrm{Sb}$	0
$\alpha_{34}\left(T\right)$	$\alpha_{AsSB}\left(T\right)$	750								

## Data Availability

Data are contained within the article.

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
