# Peer review of "Isotherm Theoretical Study of the AlxGa1−xAsySb1−y Quaternary Alloy Using the Regular Solution Approximation for Its Possible Growth via Liquid-Phase Epitaxy at Low Temperatures"

_entropy, 2022, doi:10.3390/e24121711_

Round 1

Reviewer 1 Report

1)      In the title, ‘obtaining’ word does not fit. The title should be modified to “Phase Diagrams of the AlXGa1-XAsYSb1-Y Quaternary Alloy using the Regular Solution Approximation for its Possible Growth via Liquid-Phase Epitaxy at Low Temperatures”.

2)      The language of the manuscript needs to be improved. For example, in conclusions, the last line should be “…..arsenic atomic fraction reaches a very small value……”. Correct the sentence in line 189-190.

3)      Subscripts should be used correctly. For example in page 5, line 180 and also in page 4, line 154. Correct this throughout the manuscript.

4)      Table 1 title is grammatically incorrect. It should be “Interaction parameters, fusion entropies, and fusion temperatures of binary compounds, along with interaction parameters for the ternary compounds”

5)      In few sentences, full stop should be used instead of comma. Long sentences make reading unnecessarily difficult. For example, in page 2, line 57-60.

6)      DOI in reference 3 is missing.

Author Response

Enclosed you will find the revised version of the manuscript: “Phase Diagrams of the AlxGa1-xAsySb1-y Quaternary Alloy using the Regular Solution Approximation for its Possible Growth via Liquid-Phase Epitaxy at Low Temperatures", by Erick Gastellóu, Rafael García, Ana Maria Herrera, Antonio Ramos, Godofredo García, Mario Robles, Jorge Alberto Rodríguez, Yani Dallane Ramírez, Roberto C. Carrillo, and Gustavo Alonso Hirata. We have implemented several changes in the text to answer all indications requested by the assistant editor and reviewer. We believe that their observations allowed us to improve substantially the quality of our work, and hence we manifest our gratitude to the professional job they did in the review of our manuscript.

Reviewer 2 Report

 Present reviewer has served as a materials scientist including physical metallurgy of steels and light-emitting semiconductors working at several universities and a steel company.

This paper deals with AlxGa1-xAsySb1-y quaternary compound semiconductors with x=0.0000001~0.0001 and y=0.001~0.1. They are dilutely doped compound semiconductors.  Therefore, and it has nothing to do with the aim and scope of this special issue of current high-entropy alloys. 

Even if viewed as a regular research paper, the concept of the regular solution approximation is old-fashioned. This analysis is based on the very old papers published in 1970ies. It is also pointed out that many parameters and experimental values tabulated in Fig.1 are given with no references. 

Finally, what is the technical significance of the study, if any?

Author Response

(The authors gave the same response as above.)

Reviewer 3 Report

This author presents the theoretical calculation of the phase diagrams for the AlxCa1-xAsYSb1-Y quaternary alloys through the regular solution approximation. However, several problems should be addressed before the publication of this manuscript. The review comments are shown as follows:

1.    The introduction part of the current manuscript is too concise. The author did not explain why the regular solution approximation is adopted in the present work. Currently, this model is not widely used in the thermodynamic modeling of alloys and ceramics. There are a lot of other models such as the sublattice model, quasichemical mode, and ionic liquid model. If this model is powerful for the modeling of this system, there should be much explanation on it.

2.    The literature review of this manuscript is not enough. Only 9 references in the Introduction part. More references should be added such as.

a.     Pessetto, J.R. and Stringfellow, G.B., 1983. AlxGa1− xAsySb1− y phase diagram. Journal of Crystal Growth62(1), pp.1-6.

b.    Hassan, F.E.H., Postnikov, A.V. and Pagès, O., 2010. Structural, electronic, optical and thermal properties of AlxGa1− xAsySb1− y quaternary alloys: first-principles study. Journal of alloys and compounds504(2), pp.559-565.

3.    The references cited in the present paper are too old. The author should include more references from the latest 20 years.

4.    Figure 1 is not a phase diagram.

5.    The x and y tick labels in Figure 1 should be consistent with each other (please use 1E-X or 10-X for all the tick labels).

Author Response

(The authors gave the same response as above.)

Round 2

Reviewer 2 Report

1.The authors stick to the words of "phase diagrams". Then, ordinary definition of the phase diagrams is required. In another words, the fields of Liquid (L), Solid (S) or L+S should be clearly indicated in Figs. 1 and 2. If impossible, this title should be discarded. Otherwise, this paper will be wrongly cited in the scientific databases.

2. The authors are requested to describe how they have examined the validity of their equations (1) to (12') by referencing the cited literature.

3. The authors write in the title as "possible growth via liquid-phase epitaxy".  In the Conclusion section, they state that "possible growth of layers at temperatures below 500C".  Do they assert that the epitaxy is impossible at 550C or 300C? Please show the concept of the liquid-phase epitaxy by an illustration in the manuscript.

Author Response

Comments of the second reviewer:

  1. The authors stick to the words of "phase diagrams". Then, ordinary definition of the phase diagrams is required. In another words, the fields of Liquid (L), Solid (S) or L+S should be clearly indicated in Figs. 1 and 2. If impossible, this title should be discarded. Otherwise, this paper will be wrongly cited in the scientific databases.

Answer: The phase diagram concept was changed to isotherms due to that these are horizontal lines in which transformations take place between mixtures of various components with a minimum melting point (solidification), which is lower to each of the components in its pure state. In this case, the isotherms show the relationship between the aluminum, arsenic, and antimony compositions in a gallium-rich zone as a function of different temperatures (350 °C, 400 °C, 450 °C, and 500 °C. Furthermore, as mentioned the reviewer the title was changed.

  1. The authors are requested to describe how they have examined the validity of their equations (1) to (12') by referencing the cited literature.

It is important to mention that the equations presented are based on references 15 and 16, from which the conceptual and mathematical development was done. The developed equations are valid; they have even been used in defending of a degree exam.

  1. The authors write in the title as "possible growth via liquid-phase epitaxy". In the Conclusion section, they state that "possible growth of layers at temperatures below 500C". Do they assert that the epitaxy is impossible at 550C or 300C? Please show the concept of the liquid-phase epitaxy by an illustration in the manuscript.

Answer: A new Figure to describe the liquid-phase epitaxy was included in manuscript, then using Figure 1, the LPE method was described, however it is important to mention that the possible growth of layers at low temperatures is sought in order to improve the crystalline quality material, which could be used as substrate for the obtaining of III-Nitride. Based on the reviewer's comment, the conclusion was corrected to be concise and not confuse readers.

Thank you very much for your kind comments, which have helped to substantially improve our work.

Reviewer 3 Report

No

Author Response

November 2st, 2022

Dr. Tim Li
Assistant Editor

Entropy

Reviewer 3

Manuscript ID: entropy-1932415

Dear Dr. Tim Li, and Reviewer 3

Enclosed you will find the revised version of the manuscript: “Isotherms Theoretical Study of the AlxGa1-xAsySb1-y Quaternary Alloy using the Regular Solution Approximation for its Possible Growth via Liquid-Phase Epitaxy at Low Temperatures", by Erick Gastellóu, Rafael García, Ana Maria Herrera, Antonio Ramos, Godofredo García, Mario Robles, Jorge Alberto Rodríguez, Yani Dallane Ramírez, and Roberto C. Carrillo. We have implemented several changes in the text to answer all indications requested by the assistant editor and reviewer. We believe that their observations allowed us to improve substantially the quality of our work, and hence we manifest our gratitude to the professional job they did in the review of our manuscript.

On the platform the third reviewer does not show comments, however, several corrections were made, which are marked in green in the revised manuscript, hoping to improve the quality of the manuscript. The manuscript was also proofread by a native grammar checker.

Thank you very much to Reviewer third for your kind comments, which have helped to substantially improve our work.

            Yours Sincerely

         Dr. Erick Gastellóu

[email protected]
